# Early Detection of Wild Ungulate Herbivory Cessation in Mediterranean Landscapes Using Hill Numbers as Vascular Plant Diversity Indices

José M. García del Barrio [1,*], Ramón Perea [2,3], Rafael Villafuerte-Jordán [1] and María Martínez Jaúregui [1]

1 Instituto de Ciencias Forestales, INIA-CSIC, 28040 Madrid, Spain; rafa.arfa@gmail.com (R.V.-J.); martinez.maria@inia.csic.es (M.M.J.)
2 Departamento de Sistemas y Recursos Naturales, Universidad Politécnica de Madrid, 28040 Madrid, Spain; ramon.perea@upm.es
3 Centro para la Conservación de la Biodiversidad y el Desarrollo Sostenible (CBDS), Universidad Politécnica de Madrid, 28040 Madrid, Spain
* Correspondence: jmgarcia@inia.csic.es

**Abstract:** Herbivory by ungulates is a seminal driving force in Mediterranean landscapes, where habitat diversity contributes to supporting high population densities. We investigated the influence of grazing and browsing, primarily by red deer (*Cervus elaphus*), on herbaceous and woody plant species, using a twin-plot design with herbivory exclusion. The prompt detection of herbivory cessation in vegetation was measured in multiscale plots by calculating Hill's numbers (0, 1, and 2) as diversity indices over two years. The results revealed an increased diversity gradient by habitats (Pine reforestation→Mediterranean woodland→dehesas of *Quercus* spp.) with an initial increase in overall species and herbaceous species richness in the fenced plots. Woody vegetation did not change significantly in species richness, or typical or dominant ones. In addition to the early changes detected in the richness of herbaceous species ($^0D_{her}$), medium–long term variations in woody species (presence and abundance) would represent good indicators of herbivory pressure for a diverse array of Mediterranean habitats.

**Keywords:** woody and herbaceous coverage; ungulate exclusion; biodiversity; *Cervus elaphus*; exponential of Shannon–Weaver index; inverse of Simpson concentration index; species richness

## 1. Introduction

Present-day species assemblages and landscapes are a manifestation of the driving forces that have operated from ancient times up to the present [1]. Herbivores are a fundamental component of any ecosystem and one of the main driving forces behind vegetation dynamics, exerting a strong and complex influence on their habitats as well as on local and regional biodiversity [2]. Livestock grazing and the 'overabundance' of large wild herbivores in forested areas have long been viewed as conflicting with the goals of both silviculture and forest conservation [3]. However, herbivory is also essential for maintaining habitat values in forest ecosystems and supporting biodiversity [4].

The Mediterranean Basin exemplifies a long-term use of the territory by human populations, evident in numerous examples of what are termed cultural landscapes [5–7]. In these Mediterranean landscapes, wild ungulate populations have exerted variable impacts on vegetation, which has also been influenced by the pressures of human settlements and their livestock. Wild ungulate populations have dramatically increased in Europe and other regions of the Northern Hemisphere [8–11]. These changes are primarily attributed to socio-economic shifts and land-use changes (such as the abandonment of livestock, surge in the investment of big game hunting, and increased forest cover), as well as the absence of natural predators [11–13].

Deer (Cervidae) populations have experienced significant anthropogenic impacts on their distribution over the past centuries, with conflicting effects due to conservation policies and the economic importance as a game species [14,15] and as hunting trophies [16,17]. Today, the management of these populations in extensive game estates, as well as in protected areas and natural parks, is a contentious issue [18,19]. Some concerns include competition with domestic livestock [20], disease transmission [21,22], pressure on crops and urban areas [23,24], and their significance in ecotourism projects in depopulated regions [17,25]. However, it is essential to remember that one of the main management objectives is understanding and regulating the extent of herbivory pressure on vegetation in general, and specifically on the key habitats that deer occupy [26,27]. In this context, the question arises as to what extent plant diversity indices could be the early indicators of changing wild ungulate pressure on vegetation, particularly from large herbivores like deer [28]. Previous studies have highlighted the value of certain plant taxa as early indicators of deer overabundance and, hence, over browsing, including potential threats to the regeneration and conservation of endangered plant species [10,29,30].

The list of proposed indicators for evaluating biodiversity (see, for example, [31]) and the indices used to measure the biological diversity of animal and plant groups (e.g., [32]) is as extensive as the list of researchers interested in the subject. In Mediterranean landscapes, the mosaic of land uses, along with the distribution and extent of vegetation habitats—such as forests, woodlands, scrublands, and savanna-like dehesas—tend to serve as reliable biodiversity indicators at both local and regional levels [33,34]. Additionally, the measurement of vascular plant diversity can effectively characterize the vegetation within each habitat (alpha diversity) and its variations across regional landscapes (gamma diversity) [35,36], offering a broad perspective of diversity at a regional scale.

The proposal of a diversity index continuum by Hill [37], later reinterpreted as Hill's number or the equivalent number of species [38–40], provides a framework that unifies the qualitative and quantitative aspects of diversity indices. This approach allows for the measurement of not only species richness but also evenness and dominance, all in comparable units. Hill's numbers contribute to diversity indices by utilizing the concept of the equivalent number of species. Although Hill presented these calculations as a continuum, the first three integer indices ($^0D$, $^1D$, and $^2D$) correspond to different transformations of widely used diversity indices. Specifically, $^0D$ equals S, representing species richness; $^1D$ equals the exponential of the Shannon–Weaver index; and $^2D$ equals the inverse of Simpson's concentration index [41]. In this reinterpretation, $^0D$ represents total species, $^1D$ represents typical species, and $^2D$ represents dominant species [42,43]. This continuum of values, or the relationships between them, represented by diversity profiles (see, example [44]), can serve as diversity status for the sampling units analyzed, whether at local or regional levels.

Long-term experimental exclusion designs have been widely used to monitor the effects of herbivory on vegetation. Examples include studies in Alaska [45], in Kenya [46], or in Portugal [47]. Previous research (e.g., [4,48]) has reported varying results. In a systematic review of 144 studies conducted in temperate and boreal forests, Bernes et al. [4] found a negative effect of herbivory on the abundance of overall understory vegetation and the species richness of woody understory plants, while the species richness of forbs and bryophytes responded positively. In Mediterranean woodlands and pastures, Saatkamp et al. [49] observed that vegetation recovers quickly after short-term grazing abandonment, but long-term grazing exclusion leads to a reduction in species richness, changes in vegetation structure, and alterations in soil properties.

In this context, we initiated a new long-term experiment to monitor the responses of various ecosystem components (soil, fungi [50], invertebrates, and vascular plants) to herbivory exclusion under Mediterranean mid-range conditions. We implemented a systematic twin-plot design (400 m$^2$ plots, with and without herbivore exclusion) within a fenced public game estate (~6800 ha), where red deer (*Cervus elaphus* L.) populations have been monitored for the past 30 years. The short-term objective is to detect the immediate vegetation responses (both woody and herbaceous species) to ungulate exclusion across

local habitats, including Mediterranean woodlands, pine reforestations, and *Quercus* spp. dehesas. Additionally, we aim to identify the most efficient and sensitive indices for detecting the response of each habitat and its vascular plant species assemblages to ungulate pressure cessation.

We hypothesize that a multiscale vegetation survey method, combined with quantitative data collection and the calculation of Hill numbers' diversity indices, will serve as an effective monitoring approach for detecting local ($\alpha$) and regional ($\gamma$) changes driven by herbivory cessation. In addition, we expect a different response from herbaceous vs. woody species and an overall and detectable difference depending on habitat type. Specifically, we expect a greater diversity effect of herbivory exclusion on those habitats dominated by herbaceous species.

## 2. Materials and Methods

### 2.1. Study Site

Los Quintos de Mora is a public hunting estate of 6864 hectares located in the Montes de Toledo range, Central Spain (Figure 1a). The climate is continental Mediterranean, with hot and dry summers and very variable rainfall. Over the last five years, total precipitation was 450.56 mm/year, with an average summer precipitation (June, July, and August) of 26.6 mm (diary data from the Quintos de Mora meteorological station). The soils are poor in nutrients and are acidic (pH 5.2) with a lithological substrate of quartzite and slates [51]. The property extends between two mountain ranges, from the north (SOLANA) to the south (UMBRÍA), with gentle foothills and a central plain (RAÑA) where non-permanent watercourses flow. There are three distinct vegetation types: (1) Sclerophyllous and semideciduous oak woodlands and scrublands (main species *Quercus ilex* L., *Quercus faginea* Lam., *Arbutus unedo* L., *Cistus ladanifer* L., *Erica* L. spp.) in the mountain's ranges. (2) Pine forests resulting from reforestations in the middle of the last century, with *Pinus* L. spp. (*Pinus pinea* L.in SOLANA foothills and the driest zones of the RAÑA, and *Pinus pinaster* Aiton in UMBRÍA foothills). (3) Dehesas (savannah-like systems) of *Q. ilex* and *Q. faginea* at more humid zones of the RAÑA, with a large herbaceous cover. The property is perimeter-fenced and actively managed, where the conservation of habitats and landscapes is compatible with high densities of deer populations (just over thirty-five individuals per km$^2$ [52]), and lower densities of other herbivores such as wild boar (*Sus scrofa* L.) and roe deer (*Capreolus capreolus* L.). To this end, rotating crop sheets are established, mainly in the clearer pine forest areas (around twenty per year and with a total area of approximately 250 ha). Rye, clover, and a mixture of cereals (mainly barley, oats, and wheat) are usually planted for feeding in situ or after mowing, to supplement the natural forage in pastures and scrubland.

### 2.2. Experimental Design

Based on the systematic sampling design used by the National Forest Inventory [53] (plots located at the nodes of a 1 × 1 km$^2$ grid), thirty twin plots were defined. This grid reference was used to establish the twin plots, covering all habitats of the hunting estate. Each twin consists of two plots: an open (unfenced) plot measuring 20 × 20 m$^2$, with a camera trap positioned at its center, and a closed (fenced) plot of the same size, from which deer were excluded in December 2020. The fenced plot replicates the conditions of the open plot as closely as possible (in terms of lithology, slope, orientation, and vegetation cover) within the same habitat and located between 50 m and 200 m distance. The location of the fenced plots is shown on the representation of the estate (Figure 1a). The methodology detailed in Appendix A was used to validate the experimental design of the twin plots.

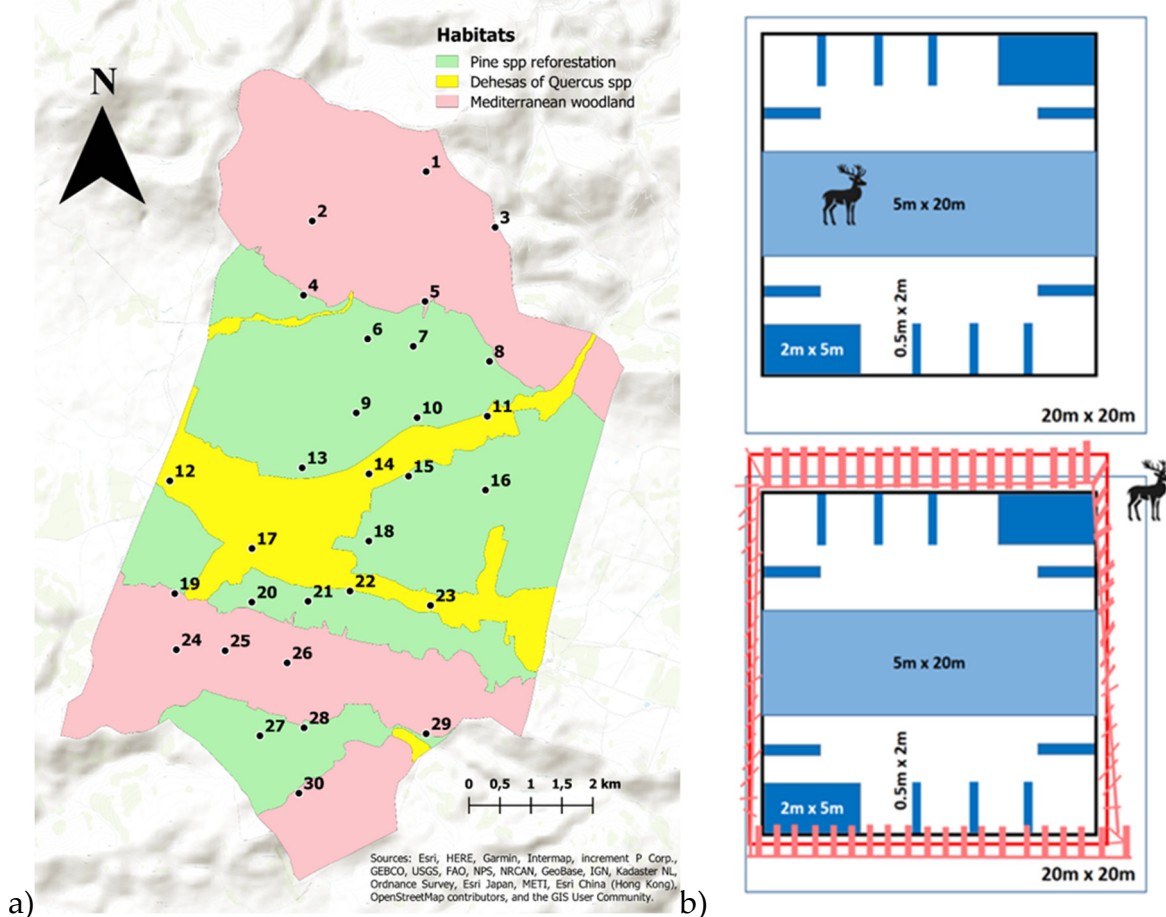

**Figure 1.** (**a**) "Los Quintos de Mora" with the main habitat's distribution and the 30 fenced plots sampled represented by circles (**b**) twin plots (open and perimeter-fenced) and the multiscale sampling design.

*2.3. Data Acquisition*

The presence and abundance of vascular plant species were sampled using a Whittaker multiscale plot design [54,55]. In this case, the plot sampling areas measured $20 \times 20$ m$^2$. The sampling period took place in mid-spring of 2021 (20 twin plots) and 2022 (10 twin plots not sampled in 2021). Each twin plot (Figure 1b) was sampled on the same day to avoid phenological gaps in species identification and growth status. In 2022, nine of the twenty twin plots sampled in 2021 were resampled (six in the Mediterranean woodland habitat—three in SOLANA and three in UMBRÍA—and three in the Dehesas of the *Quercus* spp. habitat) to check for differences in the diversity index results attributable to the sampling year.

The multiscale plot sampling followed a nested design as outlined below:

(i)     Ten subplots of $0.5 \times 2$ m$^2$, arranged equidistantly within the plot, with the outer boundary of each subplot lying on the perimeter of the plot. These subplots were used to measure the presence and abundance of all vascular plants.

(ii)    Two subplots of $2 \times 5$ m$^2$, placed in opposite corners of the plot, with their outer boundaries on the perimeter. These subplots were used to additionally measure the presence and abundance of woody plants.

(iii)   A subplot of $5 \times 20$ m$^2$ (100 m$^2$), placed in the center of the plot, without contact with any of the other subplots. This subplot was used to measure the presence and abundance of tree species and to record any vascular species not previously detected.

(iv)    Finally, a general survey of the entire $20 \times 20$ m$^2$ plot to detect any new species not previously recorded.

Species abundance was recorded according to five visually estimated cover categories: (1) punctual presence corresponding to 1% of the total subplot area, (2) less than 10% of the total subplot area, (3) between 10% and 25% of the total subplot area, (4) between 25% and 50% of the total subplot area, and (5) >50% of the total subplot area. The entire 400 m$^2$ plot was surveyed for species not found in the 1, 10, and 100 m$^2$ subplots, and an abundance rate of 0.0001% was assigned to these species. The transformation of cover categories to percent cover was applied to obtain a final quantitative species/plot matrix (see Appendix B for species list with presence and abundance data).

## 2.4. Validation of Twin Plot Experimental Design

First, we verified that our experimental design of twin plots corresponds to a set of thirty pairs in which the vegetation is significantly more similar than for any group of thirty pairs chosen at random among the sampled plots. To do this, we compared the results of two similarity indices: a qualitative index (Jaccard index) and a quantitative index (Morisita–Horn index) calculated with Estimates 9.0 [56]. The possible pairs resulting from the combination of the sixty sampled plots reach a value of 1770 pairs (Appendix A). From this dataset we randomly combined groups of thirty pairs, and the corresponding descriptive statistics for each group were calculated. This operation was repeated one thousand times, and the distribution of the similarity means of all the groups was compared with the average and standard deviation of the similarity values obtained from our set of thirty twin samples (using the "foreach" and "ggplot2" packages; R v.4.1.2).

## 3. Calculation of Local and Regional Diversity

Second, we calculated alpha (local), gamma (regional), and beta multiplicative diversity indices, using Hill numbers ($^0$D, $^1$D, and $^2$D) to determine the three equivalent numbers of species (total, typical, and dominant species, respectively). All calculations were performed using Estimates 9.0 [56]. The next step involved disaggregating the initial matrix according to (a) vegetation type (woody or herbaceous), (b) exclusion level (open or fenced), and (c) habitat type (dehesas of *Quercus* spp., Mediterranean woodland, and *Pinus* spp. reforestations). Following the systematic sampling design, we sampled 30 plots in *Pinus* spp. reforestations (15 twin plots), 20 plots in Mediterranean woodlands (10 twin plots), and 10 plots in dehesas of *Quercus* spp. (5 twin plots). Comparisons between habitat indices were also made by applying rarefaction to the minimum number of samples (n = 10).

We also calculated herbaceous and woody species gross abundance (sum of the abundance of each species vegetation type in the plot) and analyzed the differences between fenced and open plots by a t-test for means comparison. We finally compared by the same test differences between fenced and open plots in Hill numbers for local diversity ($\alpha$).

The twin plot experimental design was validated following the results outlined in Appendix A (Figure A1).

The floristic collection from Los Quintos de Mora estate comprises over 800 taxa, a noteworthy figure for such a small enclave of continental Mediterranean vegetation [57]. During our two spring sampling periods, we recorded 307 taxa across 60 multiscale sampling plots, each covering 400 m$^2$, for a total sampling area of 2.4 hectares. This means that we sampled approximately 40% of the total taxa in just 0.04% of the area. A complete list of the species found, including their frequency (number of plots) and total coverage, is provided in Appendix B (Table A1).

## 3.1. Woody and Herbaceous Species Diversity

Hill numbers for D = 0, 1, 2 at the local ($\alpha$), estate-wide ($\Upsilon$), and corresponding multiplicative dissimilarity ($\beta$) levels are presented in Table 1 for all species, as well as for woody and herbaceous species.

**Table 1.** Plant diversity (Hill numbers) for α (alpha or local diversity), Υ (gamma or landscape diversity), and β multiplicative (beta or between samples dissimilarity) in the 60 sampled plots (30 twin plots).

| | $^0$D (Species Richness) | | | $^1$D (Typical Species) | | | $^2$D (Dominant Species) | | |
|---|---|---|---|---|---|---|---|---|---|
| | α | Υ | β | α | Υ | β | α | Υ | β |
| All plant species | 45 | 307 | 6.67 | 11.67 | 31.04 | 2.66 | 6.79 | 16.96 | 2.50 |
| Woody species | 9 | 46 | 5.11 | 5.42 | 13.79 | 2.54 | 4.12 | 10.83 | 2.63 |
| Herbaceous species | 36 | 261 | 7.25 | 12.59 | 37.41 | 2.97 | 7.01 | 8.97 | 1.28 |

Comparing the three indices we observe that woody species represent about 20% of total species at plot level (α), and 15% at landscape level (Υ). These figures increase for woody typical species to 46% (α) and 44% (Υ), being the dominant species in the higher percentages of 60% (α) and 64% (Υ). The β dissimilarity values indicate a strong influence of herbaceous species on species richness, an intermediate influence on typical species figures, but a reduced influence when accounting for abundant species. The relative contributions of woody and herbaceous plants are discussed in more detail in Appendix C (Figure A2).

*3.2. Open vs. Fenced Plot Diversity*

Hill numbers for open and fenced plots showed significant differences in local species richness, considering all species ($^0$D$_{tot}$; t = −2.283; $p$ = 0.026) and herbaceous species ($^0$D$_{herb}$; t = −2.097; $p$ = 0.040). Comparing the species accumulation curves for open and fenced plots, cumulative gamma diversity becomes significant (within 95% confidence intervals) when more than six randomly selected samples are included, as shown in Figure 2.

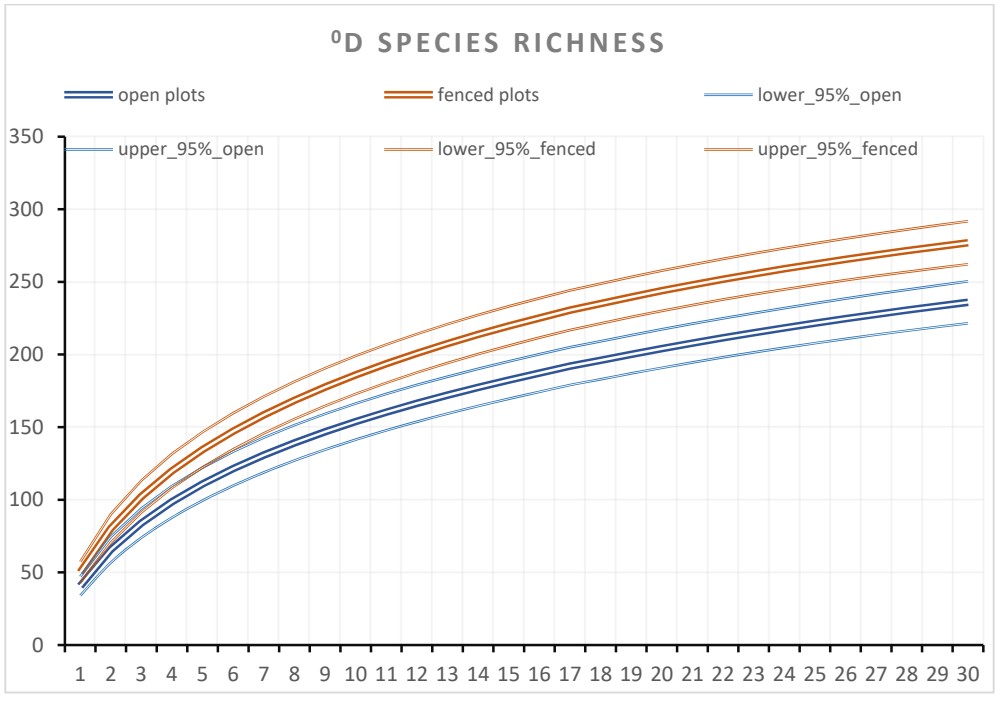

**Figure 2.** Species accumulation curves for open and fenced plots. *X* axis number of plots, *Y* axis number of species.

Regardless of species richness, herbaceous abundance (sum of the abundance of each species in the plot) showed a notable increase in fenced plots, reaching approximately 37%. A test of means applied to the herbaceous plant abundance in paired plots (alpha abundance) also indicated that herbaceous abundance was significantly higher in fenced plots (t = 9.644, $p = 1.493 \times 10^{-10}$). However, diversity indices for typical species and abundant species did not show significant differences at either the local or landscape scale.

### 3.3. Floristic Diversity of Habitats

The Hill numbers ($^0$D, $^1$D, and $^2$D) for regional diversity ($\Upsilon$), including both full sampling and rarefaction to control for differences in the number of sampled plots, are presented in Table 2. Overall, the highest diversity was found in the least extensive yet most productive habitat: *Quercus* spp. dehesas, located in the more humid areas of the RAÑA. Without applying rarefaction, the three habitats showed similar gamma diversity values ($\Upsilon$ total) for species richness but differed for typical and dominant species. The diversity gradient, based on rarefaction, follows the order: Pine reforestation→Mediterranean woodland→*Quercus* spp. dehesas for species richness and dominant species, but not for typical species (where Pine reforestation surpassed Mediterranean woodland). In any case, these figures are strongly influenced by herbaceous species, as the number of woody species remains nearly constant across habitats and indices according to our results.

**Table 2.** Regional diversity for the three habitats sampled in Los Quintos de Mora. $^0$D species richness, $^1$D typical species, $^2$D abundant species.

| | | Pine Reforestation | | Mediterranean Woodland | | Dehesas of *Quercus* spp. | |
|---|---|---|---|---|---|---|---|
| | | Total | Rarefaction | Total | Rarefaction | Total | Rarefaction |
| $^0$D | $\Upsilon$ all species | 215.0 | 143.0 | 210.0 | 153.0 | 197.0 | 197.0 |
| | $\Upsilon$ woody | 35.0 | 25.0 | 31.0 | 25.0 | 24.0 | 24.0 |
| | $\Upsilon$ herbaceous | 180.0 | 118.0 | 179.0 | 129.0 | 173.0 | 173.0 |
| $^1$D | $\Upsilon$ all species | 22.3 | 20.2 | 17.0 | 15.9 | 40.8 | 40.8 |
| | $\Upsilon$ woody | 10.2 | 9.4 | 9.9 | 9.4 | 10.8 | 10.8 |
| | $\Upsilon$ herbaceous | 23.0 | 21.2 | 37.8 | 32.3 | 40.0 | 40.0 |
| $^2$D | $\Upsilon$ all species | 11.9 | 11.0 | 9.7 | 9.2 | 18.1 | 18.1 |
| | $\Upsilon$ woody | 7.1 | 6.6 | 7.5 | 7.1 | 6.9 | 6.9 |
| | $\Upsilon$ herbaceous | 6.4 | 5.9 | 9.7 | 9.2 | 13.8 | 13.8 |

### 3.4. Other Influences on Floristic Diversity

The analysis of nine pairs of twin plots revealed a slight increase in plant coverage from 2021 to 2022, both in fenced plots (8.1%) and open plots (13.4%), resulting in an overall annual increase of 10.7%. Additionally, the equivalent number of species also increased in 2022 for both open and fenced plots, as illustrated in Figure 3. Notably, the year-on-year increases in open plots were significant, with a nearly 40% rise in typical species and more than 35% in dominant species. In fenced plots, the year-on-year increases were smaller, at 19% for typical species and 24% for dominant species. In relation with the total number of species in either open or fenced plots there was a slight increase in open plots (6.6%), and a practical equality in fenced plots.

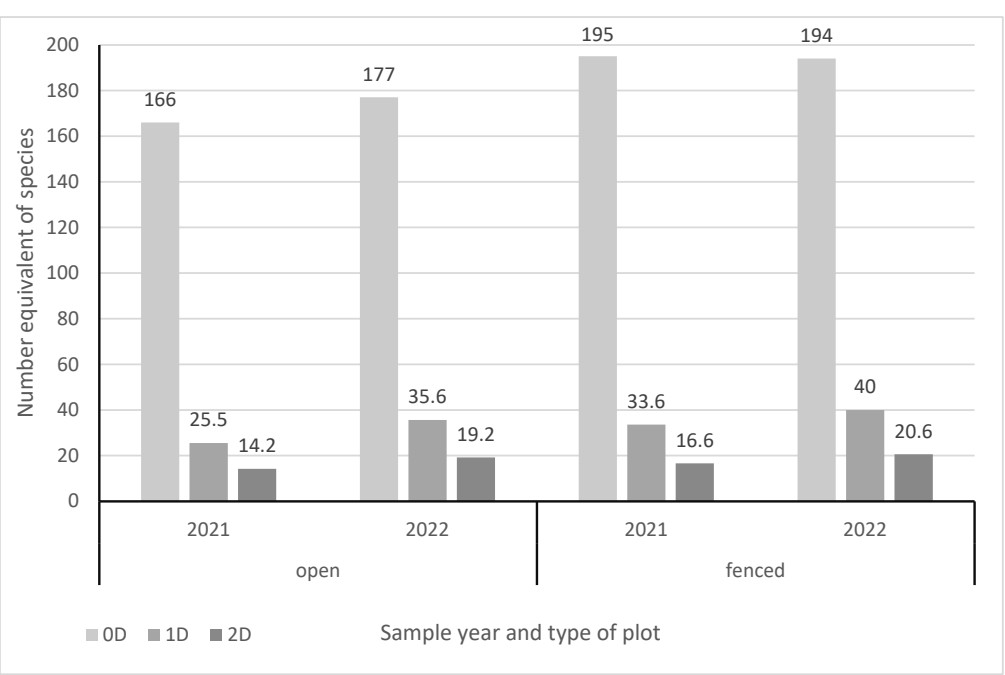

**Figure 3.** Comparison of the equivalent number of species accumulated (Y) in nine twin plots resampled in both years (2021 and 2022).

## 4. Discussion

The impact of recent wild herbivory exclusion on biodiversity was investigated using a systematic twin-plot design within a perimeter-fenced hunting estate with a high density of red deer. Vascular plant diversity, measured across three distinct habitats using multi-scale plots and calculated through Hill numbers (0, 1, and 2), was chosen as the indicator. Species richness ($^0D$), particularly herbaceous richness ($^0D_{her}$), were the first indices to display early divergence in floristic diversity between open and fenced plots, with a short-term increase observed in the fenced plots. This contrasts with findings from other studies in Mediterranean environments, where no changes in species richness were observed between grazed and ungrazed plots in the initial years following abandonment [47] (Saatkamp et al., 2018, and references therein), although changes in species composition and functional groups were noted [58]. This discrepancy may partly arise from the fact that those studies focused on grassland habitats, whereas we sampled three distinct habitat types. It may also be due to the rigorous design of our twin plots and multi-scale sampling, which effectively captures local variability.

Herbaceous species richness is the most effective indicator of early changes in the effective number of species between habitats [59]. This result was expected, as annual and biennial species (primarily herbaceous) tend to respond more quickly to disturbances—or their cessation—compared to perennial species (mainly woody), which take longer to recolonize (new species at the local scale) or show significant increases in local cover [60]. Additionally, herbaceous species richness, in Mediterranean landscapes, is significantly higher than that of woody species, even though in less managed habitats, the dominant species are woody plants that have developed long-term adaptations to grazing pressure [61]. In areas with high ungulate densities, these animals control the spread of shrubs and indirectly enhance herbaceous plant species richness by increasing the amount of light that reaches the ground [27].

Studies investigating the relationship between vegetation and grazing, whether by domestic livestock or wild herbivores, typically follow one of two approaches: (1) the cessation of grazing (herbivory exclusion) or (2) the introduction of specific grazing loads into previously ungrazed areas (e.g., [48,62]). In both cases [63], and given enough time, the intermediate disturbance hypothesis by Connell [64] appears to explain how diversity

of species varies with disturbance levels. It suggests that local species diversity peaks under intermediate levels of disturbance. In the three habitats examined in this study, we observed a gradient of disturbance intensity, with the highest disturbance occurring in the reforested pine forests of Los Quintos de Mora. These pine forests are regularly cleared and thinned to encourage natural regeneration and reduce the risk of summer fires. In such areas, a small number of deer-resistant plants (primarily non-palatable woody species) dominate the ecosystem, while less deer-resistant species tends to decline or disappear, leading to biotic homogenization [10,65]. Similarly, under low disturbance levels, as seen in Mediterranean woodland/scrubland habitats, dominant woody species tend to outcompete others, reducing overall biodiversity. For example, 30% of woody species may disappear after 30 years of herbivory exclusion [10]. Thus, the highest herbaceous species diversity is predictably found in the dehesa habitat, a flat area with higher soil moisture that supports a dense population of herbivores. Red deer limit the spread of woody species, enhancing herbaceous diversity by reducing dominance and maintaining intermediate disturbance levels.

Fencing is increasingly used as a management tool in forested landscapes [66,67], often to promote natural regeneration [68–70]. Under these conditions, it is essential to monitor changes in community composition and diversity dynamics. Long-term studies typically reveal significant shifts in species composition and abundance, with a tendency toward the homogenization of plant communities over time as herbivory is excluded (beta diversity is often used to measure this [71]). Evaluating the time required for effective regeneration, along with tracking local and regional diversity losses, will be critical in promoting grazing by wild herbivores to support the vegetation recovery process [72].

Our short-term resampling of nine twin plots over two consecutive years revealed a general trend linked to the sampling year, like observations made by other researchers in different habitats (e.g., mountain grasslands [73]). As suggested by Stuble et al. [74], we believe that community-level metrics (e.g., species richness) may reflect changes driven by environmental factors beyond those controlled in the experiment. In this case, 2022 was slightly more favorable for vegetation growth, influenced by two key environmental factors: (1) the amount and distribution of spring precipitation (March, April, and May), which in 2021 totaled 100.2 mm (only 20 mm in May), whereas in 2022 it reached 181.7 mm (but only 6.7 mm in May); and (2) the negative impact of the heavy snowfall from the Filomena storm [75] during the winter of 2021.

Our herbivory exclusion experiment, employing a multiscale method to monitor twin plots, revealed an increase in total species richness (both locally and regionally) in the exclusion plots after two growing seasons, primarily driven by an increase in herbaceous species richness. This confirms that the response to the cessation of herbivory occurs earlier in habitats most intensively used by deer, where herbaceous species dominate (dehesas of *Quercus* spp.). However, in these fenced Mediterranean estates, the relationship between vegetation type, grazing intensity, and habitat typology and extension must be systematically explored in all its relationships since, as pointed out by other authors [76,77] or recent findings of our group [78], the spatiotemporal dynamics of deer varies significantly, which implies local or punctual increases in pressure on vegetation (mainly by browsing). In these cases, the most effective indicator of this interaction is the Hill numbers that consider not only total species count but also account for species abundance by typical and dominant species. This will be crucial not only for understanding the evolution of vegetation in each habitat (woody vs. herbaceous species dominance) but also for identifying the management practices needed to maintain biodiversity.

A new resampling period, five years after exclusion and following the same methodology, will help determine the extent to which the expected increase in woody vegetation reduces diversity. It is possible that the most species-rich habitats will reach peak diversity sooner, but conversely, may experience a decline earlier as well. Understanding the temporal dynamics of diversity in each habitat, at both local and regional scales, will be essential

for developing management strategies that maintain vegetation diversity while ensuring the well-being of ungulate populations.

## 5. Conclusions

Our twin plot design enables us to compare changes in Mediterranean vegetation across the different habitats typically found in hunting estates. These changes may be attributed not only to the presence or absence of grazing and browsing by ungulate populations but also to the climatic conditions (spring precipitation) during the two consecutive springs when data were collected.

The expected divergence in vegetation between the twin plots, driven by the impact of grazing and browsing, is already detectable in the diversity index for herbaceous species richness ($^0D_{herb}$) and in the index for total species richness ($^0D_{tot}$). This is notable considering the short time interval between the fencing of the plots and the vegetation sampling.

**Author Contributions:** Conceptualization, J.M.G.d.B. and M.M.J.; methodology, J.M.G.d.B.; software, J.M.G.d.B., R.V.-J. and M.M.J.; formal analysis, J.M.G.d.B., R.P. and R.V.-J.; investigation, J.M.G.d.B., R.V.-J. and M.M.J.; writing—original draft preparation, J.M.G.d.B.; writing—review and editing, J.M.G.d.B., R.P. and M.M.J.; supervision, J.M.G.d.B. and M.M.J.; funding acquisition, M.M.J. All authors have read and agreed to the published version of the manuscript.

**Funding:** This work was supported by RTI2018-096348-R-C21 project, funded by MCIN/AEI/10.13039/501100011033 and by the FEDER "Una manera de hacer Europa". Rafael Villafuerte was supported by a FPI predoctoral contract (ref: PRE2019-091437) funded by Spanish State Research Agency.

**Acknowledgments:** We thank Ignacio Lacasa and Marina Perez from the Universidad Politécnica de Madrid for their help provided in the fieldwork. Special thanks go to C. Rodríguez-Vigal, Á. Moreno Gómez, J. Polo, A. García and other colleagues of Los Quintos de Mora for the case study information and the facilities provided in the fieldwork.

**Conflicts of Interest:** The authors declare no conflicts of interest.

## Appendix A

The Jaccard index for the 60 sampled plots, taken two by two in its 1770 combinations, presents a value of $0.24 \pm 0.09$, while for the Morisita–Horn index the value is $0.33 \pm 0.23$. The set of twin plots presents a Jaccard index of $0.43 \pm 0.10$, and for the Morisita–Horn index the value is $0.74 \pm 0.16$. The descriptive statistic of the similarity indices values for the thirty twin plots combination is significantly higher than any thirty pair plots random combination as is shown in Figure A1, which supports our choice of the twin plot group.

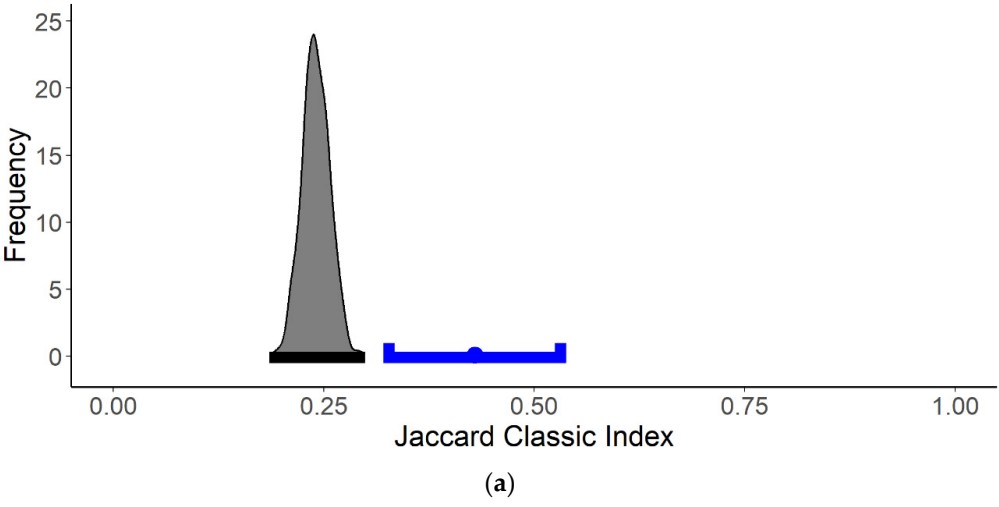

(a)

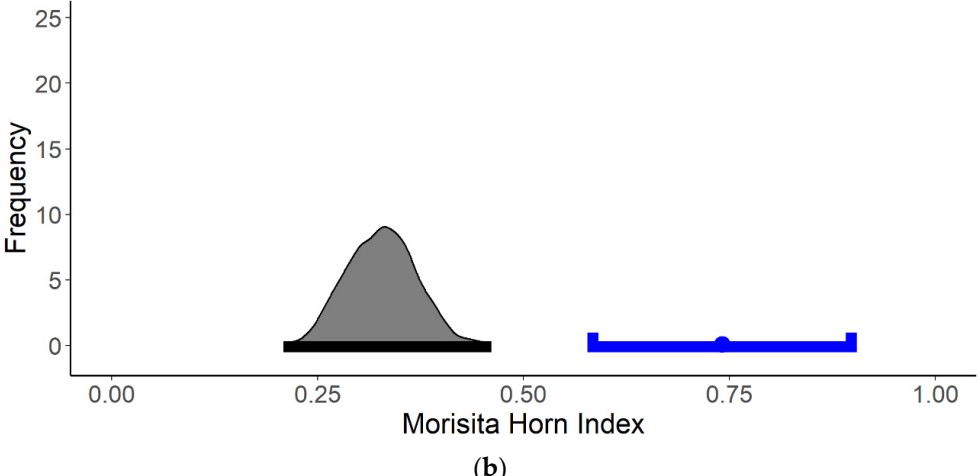

(b)

**Figure A1.** (**a**) Frequency (in gray) of similarities values of Jaccard classic for one thousand iterations of aleatory groups of 30-paired plots. In the blue color, there is the average value and standard deviation of the twin plots. (**b**) Same as (**a**) but for the Morisita–Horn index.

**Appendix B**

**Table A1.** Species list with number of appearances (count), nº plots with presence/nº total plots (frequency), and accumulated coverage (total coverage), distinguishing woody species (in italics and bold) from herbaceous species (only italics).

| Species | Count | Frequency | Coverage |
|---|---|---|---|
| ***Rosmarinus officinalis* L.** | 48 | 80.0 | 64,646 |
| ***Quercus ilex* L.** | 56 | 93.3 | 56,505 |
| ***Erica arborea* L.** | 40 | 66.7 | 55,487 |
| ***Cistus ladanifer* L.** | 47 | 78.3 | 52,960 |
| *Asphodelus albus* Mill. | 52 | 86.7 | 50,430 |
| ***Quercus faginea* Lam.** | 40 | 66.7 | 47,992 |
| ***Pinus pinea* L.** | 24 | 40.0 | 46,180 |
| ***Arbutus unedo* L.** | 21 | 35.0 | 32,047 |
| ***Pinus pinaster* Aiton** | 12 | 20.0 | 31,641 |

**Table A1.** *Cont.*

| Species | Count | Frequency | Coverage |
| --- | --- | --- | --- |
| *Phillyrea angustifolia* **L.** | 40 | 66.7 | 24,328 |
| *Thymus mastichina* **(L.) L.** | 43 | 71.7 | 14,663 |
| *Lavandula stoechas* **L.** | 22 | 36.7 | 8743 |
| *Rhamnus alaternus* **L.** | 11 | 18.3 | 6388 |
| *Tuberaria guttata* (L.) Fourr. | 50 | 83.3 | 5809 |
| *Trifolium cherleri* L. | 28 | 46.7 | 4565 |
| *Brachypodium distachyon* (L.) P. Beauv. | 27 | 45.0 | 4501 |
| *Daphne gnidium* **L.** | 25 | 41.7 | 4085 |
| *Vulpia ciliata* Dumort. | 53 | 88.3 | 3925 |
| *Vulpia myuros* (L.) C.C.Gmel. | 53 | 88.3 | 3575 |
| *Acer monspessulanum* **L.** | 5 | 8.3 | 3524 |
| *Celtica gigantea* (Link) F.M. Vázquez & Barkworth | 5 | 8.3 | 3365 |
| *Asterolinon linum-stellatum* (L.) Duby | 48 | 80.0 | 3121 |
| *Sanguisorba hybrida* (L.) Font Quer | 45 | 75.0 | 2980 |
| *Carlina racemosa* L. | 28 | 46.7 | 2956 |
| *Trifolium stellatum* L. | 14 | 23.3 | 2546 |
| *Genista hirsuta* **Vahl** | 4 | 6.7 | 2460 |
| *Bromus sterilis* L. | 17 | 28.3 | 2456 |
| *Hypochaeris glabra* L. | 43 | 71.7 | 2451 |
| *Plantago bellardii* All. | 11 | 18.3 | 2291 |
| *Pistacia terebinthus* **L.** | 12 | 20.0 | 2183 |
| *Quercus pyrenaica* **Willd.** | 5 | 8.3 | 2168 |
| *Asphodelus aestivus* Brot. | 5 | 8.3 | 2023 |
| *Bromus madritensis* L. | 35 | 58.3 | 1810 |
| *Tolpis umbellata* Bertol. | 29 | 48.3 | 1807 |
| *Taraxacum marginellum* H. Lindb. | 34 | 56.7 | 1790 |
| *Anacyclus clavatus* (Desf.) Pers. | 28 | 46.7 | 1699 |
| *Aira caryophyllea* L. | 36 | 60.0 | 1686 |
| *Cerastium glomeratum* Thuill. | 32 | 53.3 | 1674 |
| *Galium aparine* L. | 32 | 53.3 | 1654 |
| *Helianthemum apenninum* **(L.) Mill.** | 8 | 13.3 | 1597 |
| *Andryala integrifolia* L. | 32 | 53.3 | 1529 |
| *Cynosurus echinatus* L. | 29 | 48.3 | 1500 |
| *Erodium botrys* (Cav.) Bertol. | 21 | 35.0 | 1447 |
| *Teucrium chamaedrys* **L.** | 13 | 21.7 | 1444 |
| *Erica umbellata Loefl.* **ex L.** | 4 | 6.7 | 1432 |
| *Briza minor* L. | 24 | 40.0 | 1373 |
| *Erica tetralix* **L.** | 3 | 5.0 | 1270 |
| *Avena barbata* Pott ex Link | 26 | 43.3 | 1259 |
| *Bromus hordeaceus* L. | 12 | 20.0 | 1247 |

**Table A1.** *Cont.*

| Species | Count | Frequency | Coverage |
|---|---|---|---|
| *Dactylis glomerata* L. | 28 | 46.7 | 1231 |
| *Taeniatherum caput-medusae* (L.) Nevski | 24 | 40.0 | 1227 |
| ***Halimium ocymoides* (Lam.) Willk.** | 5 | 8.3 | 1153 |
| *Urginea maritima* (L.) Baker | 5 | 8.3 | 1119 |
| ***Lonicera etrusca* Santi** | 6 | 10.0 | 1108 |
| *Hypochaeris radicata* L. | 18 | 30.0 | 1047 |
| *Linum trigynum* L. | 16 | 26.7 | 1041 |
| *Briza maxima* L. | 19 | 31.7 | 1004 |
| ***Teucrium fruticans* L.** | 3 | 5.0 | 944 |
| *Arrhenatherum elatius* (L.) P. Beauv. ex J. Presl & C. Presl | 24 | 40.0 | 942 |
| ***Cistus salviifolius* L.** | 5 | 8.3 | 935 |
| ***Cistus populifolius* L.** | 5 | 8.3 | 934 |
| *Trifolium angustifolium* L. | 22 | 36.7 | 921 |
| *Filago pyramidata* L. | 26 | 43.3 | 891 |
| *Crepis vesicaria* L. | 19 | 31.7 | 888 |
| ***Crataegus monogyna* Jacq.** | 4 | 6.7 | 867 |
| *Plantago lagopus* L. | 14 | 23.3 | 840 |
| *Trifolium arvense* L. | 11 | 18.3 | 775 |
| *Aristolochia pistolochia* L. | 25 | 41.7 | 740 |
| *Taraxacum obovatum* (Willd.) DC. | 21 | 35.0 | 699 |
| *Eryngium campestre* L. | 9 | 15.0 | 682 |
| ***Asparagus acutifolius* L.** | 4 | 6.7 | 668 |
| *Lotus corniculatus* L. | 15 | 25.0 | 642 |
| ***Dorycnium pentaphyllum* Scop.** | 5 | 8.3 | 621 |
| *Euphorbia exigua* L. | 27 | 45.0 | 620 |
| ***Adenocarpus telonensis* (Loisel.) DC.** | 3 | 5.0 | 603 |
| *Linum bienne* Miller | 21 | 35.0 | 596 |
| *Anthoxanthum aristatum* Boiss. | 12 | 20.0 | 530 |
| *Trifolium campestre* Schreb. | 19 | 31.7 | 528 |
| *Rumex bucephalophorus* L. | 9 | 15.0 | 526 |
| *Ranunculus paludosus* Poir. | 22 | 36.7 | 523 |
| *Sherardia arvensis* L. | 12 | 20.0 | 510 |
| *Centranthus calcitrapae* (L.) Dufr. | 18 | 30.0 | 504 |
| *Lolium rigidum* Gaudin | 9 | 15.0 | 491 |
| *Plantago lanceolata* L. | 10 | 16.7 | 486 |
| *Rubia peregrina* L. | 11 | 18.3 | 485 |
| *Erodium cicutarium* (L.) L'Her. | 16 | 26.7 | 481 |
| ***Genista tournefortii* Spach** | 1 | 1.7 | 470 |
| ***Flueggea tinctoria* (L.) G.L. Webster** | 1 | 1.7 | 467 |
| *Jasione montana* L. | 28 | 46.7 | 467 |
| *Geranium molle* L. | 12 | 20.0 | 456 |

**Table A1.** *Cont.*

| Species | Count | Frequency | Coverage |
|---|---|---|---|
| *Airopsis tenella* (Cav.) Asch. & Graebn | 5 | 8.3 | 440 |
| ***Rhamnus saxatilis* Jacq.** | 3 | 5.0 | 433 |
| *Crucianella angustifolia* L. | 15 | 25.0 | 426 |
| ***Juniperus oxycedrus* L.** | 1 | 1.7 | 417 |
| ***Erica scoparia* L.** | 3 | 5.0 | 410 |
| *Ornithopus compressus* L. | 12 | 20.0 | 405 |
| *Anemone palmata* L. | 16 | 26.7 | 400 |
| *Petrorhagia nanteuilii* (Burnat) P.W. Ball & Heywood | 22 | 36.7 | 398 |
| *Coronilla repanda* (Poir.) Guss. | 4 | 6.7 | 390 |
| *Thapsia villosa* L. | 20 | 33.3 | 384 |
| *Filago pygmaea* L. | 15 | 25.0 | 356 |
| *Poa annua* L. | 6 | 10.0 | 350 |
| *Brachypodium sylvaticum* (Huds.) P. Beauv. | 9 | 15.0 | 341 |
| ***Rhamnus lycioides* L.** | 2 | 3.3 | 334 |
| ***Rosa canina* L.** | 2 | 3.3 | 334 |
| ***Rubus ulmifolius* Schott** | 2 | 3.3 | 334 |
| *Anthemis arvensis* L. | 8 | 13.3 | 327 |
| *Carlina corymbosa* L. | 8 | 13.3 | 327 |
| *Orobanche latisquama* (F.W. Schultz) Batt. | 22 | 36.7 | 321 |
| *Paeonia broteri* Boiss. & Reut. | 18 | 30.0 | 309 |
| *Anagallis arvensis* L. | 7 | 11.7 | 291 |
| *Geum sylvaticum* Pourr. | 4 | 6.7 | 286 |
| *Leontodon tuberosus* L. | 8 | 13.3 | 280 |
| *Tragopogon porrifolius* L. | 15 | 25.0 | 280 |
| *Vicia benghalensis* L. | 4 | 6.7 | 270 |
| *Galium parisiense* L. | 9 | 15.0 | 265 |
| *Trifolium striatum* L. | 5 | 8.3 | 260 |
| *Micropyrum tenellum* (L.) Link | 5 | 8.3 | 235 |
| *Ranunculus bulbosus* L. | 7 | 11.7 | 230 |
| *Myosotis discolor* Pers. | 9 | 15.0 | 220 |
| *Trifolium scabrum* L. | 4 | 6.7 | 220 |
| *Trifolium hirtum* All. | 5 | 8.3 | 210 |
| *Bartsia trixago* L. | 17 | 28.3 | 202 |
| *Poa trivialis* L. | 2 | 3.3 | 200 |
| *Tamus communis* L. | 11 | 18.3 | 191 |
| *Polycarpon tetraphyllum* (L.) L. | 3 | 5.0 | 190 |
| *Ornithogalum baeticum* Boiss | 9 | 15.0 | 187 |
| *Crupina vulgaris* Pers. ex Cass. | 4 | 6.7 | 186 |
| *Cruciata glabra* (L.) Ehrend. | 8 | 13.3 | 185 |
| *Scilla verna* Huds. | 2 | 3.3 | 181 |
| *Stellaria media* (L.) Vill. | 3 | 5.0 | 181 |

**Table A1.** *Cont.*

| Species | Count | Frequency | Coverage |
|---|---|---|---|
| *Cynosurus elegans* Desf. | 2 | 3.3 | 180 |
| *Anthyllis vulneraria* L. | 6 | 10.0 | 167 |
| ***Cytisus scoparius* (L.) Link** | 1 | 1.7 | 167 |
| ***Genista falcata* Brot.** | 1 | 1.7 | 167 |
| ***Lavandula pedunculata* (Mill.) Cav.** | 1 | 1.7 | 167 |
| ***Prunus spinosa* L.** | 1 | 1.7 | 167 |
| *Rumex acetosella* L. | 17 | 28.3 | 167 |
| *Crepis capillaris* (L.) Wallr. | 5 | 8.3 | 160 |
| *Filago carpetana* (Lange) Chrtek & Holub | 8 | 13.3 | 160 |
| *Torilis elongata* (Hoffmanns. & Link) Samp. | 4 | 6.7 | 156 |
| *Lathyrus cicera* L. | 6 | 10.0 | 152 |
| *Alyssum granatense* Boiss. & Reut. | 5 | 8.3 | 151 |
| *Trifolium subterraneum* L. | 2 | 3.3 | 150 |
| *Valerianella coronata* (L.) DC. | 2 | 3.3 | 150 |
| *Carduus pycnocephalus* L. | 11 | 18.3 | 145 |
| *Pilosella officinarum* F.W. Sch. & Sch. Bip. | 5 | 8.3 | 145 |
| *Aegilops triuncialis* L. | 5 | 8.3 | 137 |
| *Spiranthes aestivalis* (Poir.) Rich. | 6 | 10.0 | 130 |
| *Pimpinella villosa* Schousb. | 3 | 5.0 | 125 |
| *Biscutella auriculata* L. | 3 | 5.0 | 122 |
| *Vicia angustifolia* L. | 6 | 10.0 | 121 |
| *Arenaria montana* L. | 3 | 5.0 | 120 |
| *Paronychia argentea* Lam. | 3 | 5.0 | 116 |
| *Plantago coronopus* L. | 3 | 5.0 | 115 |
| *Silene gallica* L. | 11 | 18.3 | 113 |
| *Euphorbia helioscopia* L. | 5 | 8.3 | 111 |
| *Rhaponticum coniferum* (L.) Greuter | 3 | 5.0 | 110 |
| *Senecio vulgaris* L. | 7 | 11.7 | 107 |
| *Centaurium maritimum* (L.) Fritsch ex Janch. | 4 | 6.7 | 105 |
| *Aegilops geniculata* Roth | 4 | 6.7 | 102 |
| *Centaurea melitensis* L. | 3 | 5.0 | 101 |
| *Poa bulbosa* L. | 8 | 13.3 | 101 |
| *Omphalodes linifolia* (L.) Moench | 5 | 8.3 | 92 |
| *Andryala laxiflora* DC. | 4 | 6.7 | 91 |
| *Neotinea maculata* (Desf.) Stearn | 5 | 8.3 | 91 |
| *Viola riviniana* Rchb. | 3 | 5.0 | 90 |
| *Aphanes cornucopioides* Lag. | 6 | 10.0 | 86 |
| ***Helichrysum stoechas* (L.) Moench** | 2 | 3.3 | 86 |
| *Urospermum picroides* (L.) F.W. Schmidt | 2 | 3.3 | 86 |
| ***Teucrium capitatum* L.** | 1 | 1.7 | 83 |
| *Conopodium majus* (Gouan) Loret | 3 | 5.0 | 81 |

**Table A1.** *Cont.*

| Species | Count | Frequency | Coverage |
|---|---|---|---|
| *Filago lutescens* Jord. | 8 | 13.3 | 81 |
| *Gladiolus communis* L. | 9 | 15.0 | 81 |
| *Sisymbrium irio* L. | 3 | 5.0 | 81 |
| *Geranium columbinum* L. | 3 | 5.0 | 80 |
| *Phlomis lychnitis* L. | 3 | 5.0 | 80 |
| *Ranunculus gramineus* L. | 2 | 3.3 | 80 |
| *Parentucellia latifolia* (L.) Caruel | 7 | 11.7 | 77 |
| *Silene cretica* L. | 5 | 8.3 | 76 |
| *Sonchus asper* (L.) Hill | 4 | 6.7 | 76 |
| *Bituminaria bituminosa* (L.) C.H. Stirt. | 5 | 8.3 | 72 |
| *Aira cupaniana* Guss. | 5 | 8.3 | 71 |
| *Crepis nicaeensis* Pers. | 8 | 13.3 | 71 |
| *Euphorbia falcata* L. | 3 | 5.0 | 71 |
| *Sesamoides purpurascens* (L.) G. López | 4 | 6.7 | 71 |
| *Teesdalia coronopifolia* (J.P. Bergeret) Thell. | 3 | 5.0 | 71 |
| *Filago gallica* L. | 2 | 3.3 | 70 |
| *Misopates orontium* (L.) Raf. | 7 | 11.7 | 70 |
| *Vicia cracca* L. | 2 | 3.3 | 70 |
| *Polygala microphylla* L. | 3 | 5.0 | 62 |
| *Capsella bursa-pastoris* (L.) Medik. | 3 | 5.0 | 61 |
| *Geranium lucidum* L. | 2 | 3.3 | 61 |
| *Klasea integrifolia* (Vahl) Greuter | 2 | 3.3 | 61 |
| *Neatostema apulum* (L.) I.M. Johnst. | 3 | 5.0 | 61 |
| *Torilis arvensis* (Huds.) Link | 5 | 8.3 | 61 |
| *Biscutella valentina* (Loefl. ex L.) Heywood | 2 | 3.3 | 60 |
| *Carex divulsa* Stokes | 2 | 3.3 | 60 |
| *Erophaca baetica* (L.) Boiss. | 3 | 5.0 | 60 |
| *Orobanche ramosa* L. | 2 | 3.3 | 60 |
| *Serapias lingua* L. | 2 | 3.3 | 60 |
| *Cytinus hypocistis* (L.) L. | 6 | 10.0 | 57 |
| *Narcissus triandrus* L. | 6 | 10.0 | 55 |
| *Orchis morio* L. | 3 | 5.0 | 52 |
| *Senecio jacobaea* L. | 3 | 5.0 | 52 |
| *Campanula rapunculus* L. | 2 | 3.3 | 51 |
| *Allium massaesylum* Batt. & Trab. | 1 | 1.7 | 50 |
| *Carex flacca* Schreb. | 1 | 1.7 | 50 |
| *Hordeum murinum* L. | 3 | 5.0 | 50 |
| *Muscari comosum* (L.) Mill. | 1 | 1.7 | 50 |
| *Ononis reclinata* L. | 2 | 3.3 | 50 |
| *Ononis spinosa* L. | 1 | 1.7 | 50 |
| *Sonchus oleraceus* L. | 1 | 1.7 | 50 |

**Table A1.** *Cont.*

| Species | Count | Frequency | Coverage |
|---|---|---|---|
| *Dipcadi serotinum* (L.) Medik. | 4 | 6.7 | 41 |
| *Vulpia bromoides* (L.) Gray | 3 | 5.0 | 41 |
| *Brassica barrelieri* (L.) Janka | 3 | 5.0 | 40 |
| *Holcus lanatus* L. | 2 | 3.3 | 40 |
| *Scandix pecten-veneris* L. | 2 | 3.3 | 40 |
| *Raphanus raphanistrum* L. | 8 | 13.3 | 34 |
| *Filago minima* (Sm.) Pers. | 2 | 3.3 | 31 |
| *Lactuca viminea* (L.) J. Presl & C. Presl | 4 | 6.7 | 31 |
| *Leontodon saxatilis* Lam. | 3 | 5.0 | 31 |
| *Medicago minima* (L.) L. | 4 | 6.7 | 31 |
| *Vicia sativa* L. | 3 | 5.0 | 31 |
| *Bupleurum baldense* Turra | 2 | 3.3 | 30 |
| *Sisymbrium orientale* L. | 3 | 5.0 | 30 |
| *Stachys arvensis* (L.) L. | 1 | 1.7 | 30 |
| *Viola kitaibeliana* Schult. | 3 | 5.0 | 30 |
| *Cephalanthera longifolia* (L.) Fritsch | 4 | 6.7 | 26 |
| *Arabis stenocarpa* Boiss. & Reut. | 2 | 3.3 | 25 |
| *Carex distachya* Desf. | 4 | 6.7 | 22 |
| *Papaver dubium* L. | 3 | 5.0 | 21 |
| *Trifolium glomeratum* L. | 2 | 3.3 | 21 |
| *Cerastium semidecandrum* L. | 1 | 1.7 | 20 |
| *Dianthus toletanus* Boiss. & Reut. | 1 | 1.7 | 20 |
| *Erodium moschatum* (L.) L´Hér. | 1 | 1.7 | 20 |
| *Filago arvensis* L. | 1 | 1.7 | 20 |
| *Geranium purpureum* Vill. | 2 | 3.3 | 20 |
| *Linum narbonense* L. | 1 | 1.7 | 20 |
| *Petrorhagia prolifera* (L.) P.W. Ball & Heywood | 3 | 5.0 | 20 |
| *Valerianella microcarpa* Loisel. | 1 | 1.7 | 20 |
| ***Ruscus aculeatus* L.** | 1 | 1.7 | 17 |
| *Scirpoides holoschoenus* (L.) Soják | 2 | 3.3 | 15 |
| *Vincetoxicum nigrum* (L.) Moench | 2 | 3.3 | 15 |
| *Cynoglossum creticum* Mill. | 4 | 6.7 | 12 |
| *Iberis ciliata* All. | 3 | 5.0 | 12 |
| *Papaver rhoeas* L. | 3 | 5.0 | 12 |
| *Festuca ampla* Hack. | 2 | 3.3 | 11 |
| *Saxifraga granulata* L. | 2 | 3.3 | 11 |
| *Cirsium arvense* (L.) Scop. | 1 | 1.7 | 10 |
| *Coronilla juncea* L. | 1 | 1.7 | 10 |
| *Filipendula vulgaris* Moench | 1 | 1.7 | 10 |
| *Galium verum* L. | 1 | 1.7 | 10 |
| ***Genista florida* L.** | 1 | 1.7 | 10 |

**Table A1.** *Cont.*

| Species | Count | Frequency | Coverage |
|---|---|---|---|
| *Lathyrus nudicaulis* (Willk.) Amo | 1 | 1.7 | 10 |
| *Linaria amethystea* (Vent.) Hoffmanns. & Link | 1 | 1.7 | 10 |
| *Linum strictum* L. | 1 | 1.7 | 10 |
| *Orchis mascula* L. | 1 | 1.7 | 10 |
| *Pulicaria arabica* (L.) Cass. | 1 | 1.7 | 10 |
| *Rhagadiolus edulis* Gaertn. | 1 | 1.7 | 10 |
| *Rumex pulcher* L. | 1 | 1.7 | 10 |
| *Sedum album* L. | 1 | 1.7 | 10 |
| *Senecio lividus* L. | 2 | 3.3 | 10 |
| *Stipa lagascae* Roem. & Schult. | 1 | 1.7 | 10 |
| *Trifolium gemellum* Pourr. ex Willd. | 1 | 1.7 | 10 |
| *Festuca paniculata* (L.) Schinz & Thell. | 2 | 3.3 | 6 |
| *Vincetoxicum hirundinaria* Medik. | 2 | 3.3 | 6 |
| *Avena byzantina* K. Koch | 1 | 1.7 | 5 |
| *Centaurea ornata* Willd. | 1 | 1.7 | 5 |
| *Cynodon dactylon* (L.) Pers. | 1 | 1.7 | 5 |
| *Cynosurus cristatus* L. | 1 | 1.7 | 5 |
| *Epipactis helleborine* (L.) Crantz | 1 | 1.7 | 5 |
| *Foeniculum vulgare* Mill. | 1 | 1.7 | 5 |
| *Lupinus micranthus* Guss. | 1 | 1.7 | 5 |
| *Podospermum laciniatum* (L.) DC. | 1 | 1.7 | 5 |
| *Reseda media* Lag. | 1 | 1.7 | 5 |
| *Cynoglossum cheirifolium* L. | 4 | 6.7 | 4 |
| *Bromus rubens* L. | 3 | 5.0 | 3 |
| ***Sorbus torminalis* (L.) Crantz** | 1 | 1.7 | 3 |
| ***Teucrium oxylepis* Font Quer** | 1 | 1.7 | 3 |
| *Campanula lusitanica* L. | 2 | 3.3 | 2 |
| *Centaurium erythraea* Rafn | 2 | 3.3 | 2 |
| *Linaria spartea* (L.) Chaz. | 2 | 3.3 | 2 |
| *Narcissus bulbocodium* L. | 2 | 3.3 | 2 |
| *Rumex conglomeratus* Murray | 2 | 3.3 | 2 |
| *Agrostemma githago* L. | 1 | 1.7 | 1 |
| *Allium scorzonerifolium* Desf. ex DC. | 1 | 1.7 | 1 |
| *Anagallis monelli* L. | 1 | 1.7 | 1 |
| *Aphyllanthes monspeliensis* L. | 1 | 1.7 | 1 |
| *Arenaria conimbricensis* Brot. | 1 | 1.7 | 1 |
| *Aristolochia paucinervis* Pomel | 1 | 1.7 | 1 |
| *Calendula arvensis* (Vaill.) L. | 1 | 1.7 | 1 |
| *Cardamine hirsuta* L. | 1 | 1.7 | 1 |
| *Hedypnois rhagadioloides* (L.) F.W. Schmidt | 1 | 1.7 | 1 |
| *Hypericum perforatum* L. | 1 | 1.7 | 1 |

**Table A1.** *Cont.*

| Species | Count | Frequency | Coverage |
|---|---|---|---|
| *Lactuca tenerrima* Pourr. | 1 | 1.7 | 1 |
| *Lactuca virosa* L. | 1 | 1.7 | 1 |
| *Lepidium heterophyllum* Benth. | 1 | 1.7 | 1 |
| *Linaria aeruginea* (Gouan) Cav. | 1 | 1.7 | 1 |
| *Linaria arvensis* (L.) Desf. | 1 | 1.7 | 1 |
| *Medicago polymorpha* L. | 1 | 1.7 | 1 |
| *Medicago sativa* L. | 1 | 1.7 | 1 |
| *Myosotis laxa* Lehm. | 1 | 1.7 | 1 |
| *Orchis coriophora* L. | 1 | 1.7 | 1 |
| *Papaver argemone* L. | 1 | 1.7 | 1 |
| *Sedum brevifolium* DC. | 1 | 1.7 | 1 |
| *Spergula arvensis* L. | 1 | 1.7 | 1 |

**Appendix C**

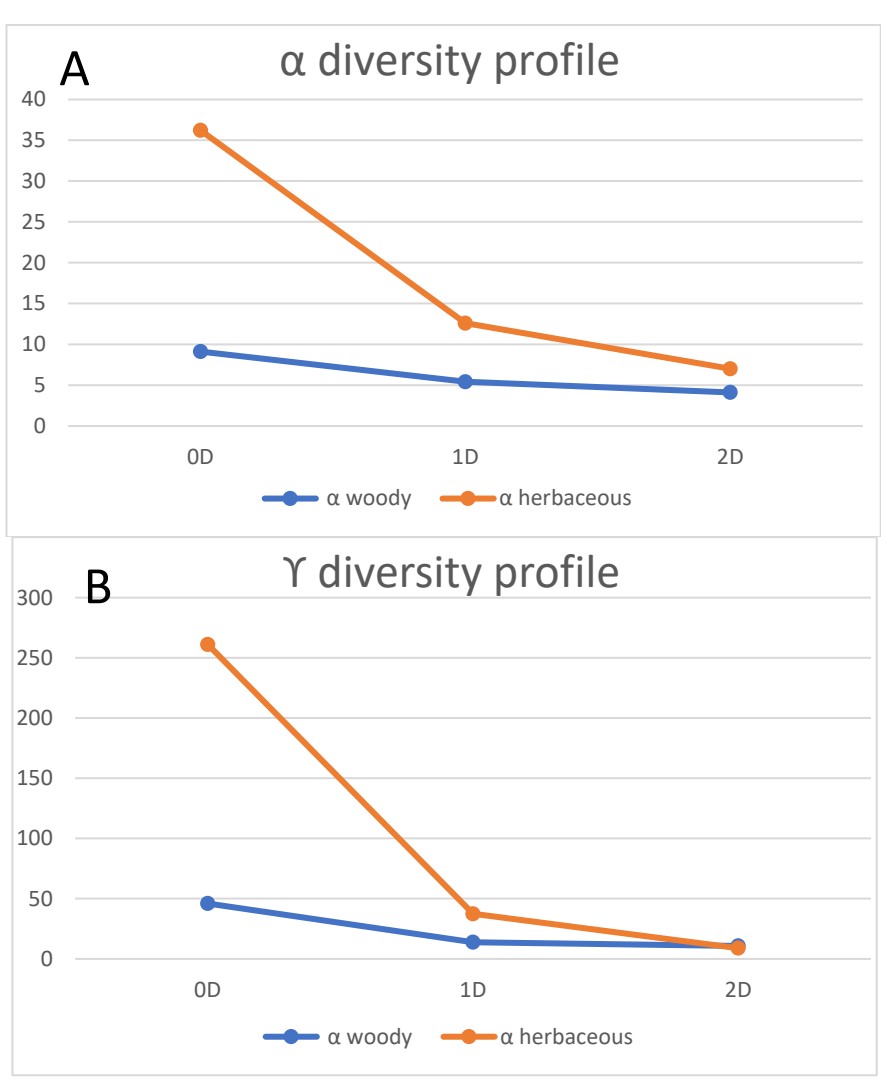

**Figure A2.** *Cont.*

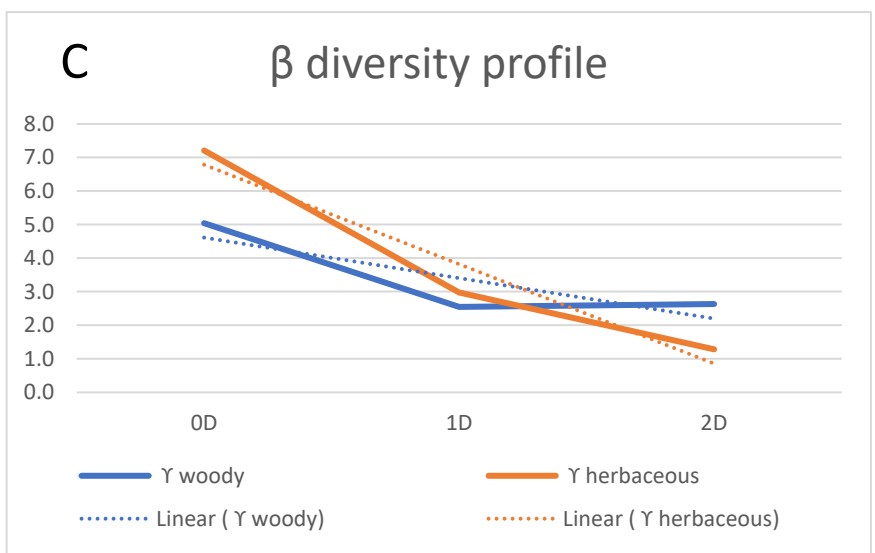

**Figure A2.** Diversity profiles for (**A**) alpha, (**B**) gamma, and (**C**) beta diversity (including the linear trend in dashed line) distinguishing between herbaceous and woody species.

Significantly, the beta diversity profile (Figure A2C) shows more heterogeneity in the herbaceous richness between plots but also great homogeneity in relation to abundant herbaceous species, as indicated by the greater slope of the trend line that begins at the highest values in $^0$D and ends at the lowest ones for $^2$D. The slope of the line for the beta profile of woody species shows less heterogeneity for $^0$D between plots (5 versus 7 for herbaceous ones) but also that there are several combinations of abundant species ($^2$D) that maintain beta values near to 3 (2.63 versus 1.28 for herbaceous ones).

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
