# Peer review of "Early Detection of Wild Ungulate Herbivory Cessation in Mediterranean Landscapes Using Hill Numbers as Vascular Plant Diversity Indices"

_land, doi:10.3390/land13122006_

Round 1
Reviewer 1 Report
Comments and Suggestions for Authors
The study presents some of the results of a long-term experiment designed to monitor the responses of various ecosystem components to herbivory exclusion under Mediterranean mid-range conditions, focusing on the immediate vegetation responses (both woody and herbaceous species) to ungulate exclusion across local habitats. The study is well written and provides a thorough presentation of the research question and methodological procedures applied. I have the a few minor comments:
i) Regarding the hypothesis. As presented, it seems to be very general and qualitative in nature, and perhaps seems to be rather a statement of assumptions rather than a hypothesis per se. In this regard, perhaps it could be articulated in a more mechanistic or process-oriented manner. As an example, given the existing literature and previous knowledge, would it be fair to expect a different response of herbaceous vs. woody species? Would this be expected only in species richness or abundance/cover? Regarding the expected differences, as stated, what would be their orientation, e.g. are woody species expected to be less/more diverse than herbaceous species and why? This would strengthen the article and allow a clearer understanding of the study's rationale. Some of this information is actually present in the discussion section in lines 286 to 296. It would seem appropriate to frame the study design and hypothesis based on this information.
ii) A related but different concern is the fact that the comparisons are focused on either a) vegetation type, b) exclusion level and c) habitat type. In this regard, the use of conventional analysis of variance designs could perhaps allow the estimation of the interaction terms across these factors. I understand this was not the original intent of the experimental design, and perhaps the number of replicates does not allow such experimental design testing. Hence, it would be interesting for the authors to analyse the relevance or potential importance of these interactions in the discussion section.
iii) Regarding figures 2 and 3, I would suggest the use of thicker axis and grid lines, as well as including the axis titles in addition to the axis labels. While the colour palettes selected seem adequate for colour blind readers, I would have sugested a monochrome design. Particularly for Figure 3
Author Response
We are sincerely grateful for the comments of both reviewers, as they have provided a comprehensive overview and a certainty in describing the gaps in the text that we appreciate.
Some of their comments focus directly on the methodological/descriptive nature of the manuscript. This duality has been present in the dialogue between the authors and I suppose we have not been able to resolve it without generating doubts about it. In any case, and with the intention of following their advice, we have introduced changes in the abstract (new lines 21-24), the statement of the hypotheses (lines 108-111) and the end of the discussion (lines 347-354) with the intention of improving the reading of the manuscript.
In response to your specific comments, we can point out:
- i) Regarding the hypothesis. We have made the changes already mentioned in the hypothesis, following your indications and the content of the discussion paragraph indicated.
- ii) The suggestion of an experimental design that takes into account the interaction of the variables (vegetation type, exclusion level and habitat type) to address a conventional analysis of variance, will be taken into account for the next sampling period, since the multi-scale design can allow different combinations of the smaller plots with a clearer orientation towards the detection of interactions.
iii) Regarding figures 2 and 3. We have followed most of your indications (thicker axis lines, and axis titles and monochrome design in Figure 3) although the axis titles remain in figure 2 caption (we think it looks cleaner this way).
Reviewer 2 Report
Comments and Suggestions for Authors
Dear Authors,
I found your study well-planned and interesting focusing on an important practical issue from both nature conservation and game management point of view. The manuscript is generally well-structured and in case of Introduction and Discussion it is easy to follow.
My two main problems were related to 1) the clear understanding of the validation of the method (the exact meaning of it and the way it was performed), 2) the description of the data analyses and pairing some of the Method section parts with the related texts in the Results.
I recommend clarifying those parts in the manuscript for an easy understanding of your messages. You can find my detailed comments in the uploaded pdf file.
Best wishes,

Author Response
We are sincerely grateful for the comments of both reviewers, as they have provided a comprehensive overview and a certainty in describing the gaps in the text that we appreciate.
Some of their comments focus directly on the methodological/descriptive nature of the manuscript. This duality has been present in the dialogue between the authors and I suppose we have not been able to resolve it without generating doubts about it. In any case, and with the intention of following their advice, we have introduced changes in the abstract (new lines 21-24), the statement of the hypotheses (lines 108-111) and the end of the discussion (lines 347-354) with the intention of improving the reading of the manuscript.
In response to your specific comments, we can also point out:
First, we would like to thank the reviewer again for his/her detailed review of the text in the attached PDF. In relation to the two main problems that he/she exposes we agree that our previous explanation about twin plot design validation was not as clear as we had expected. In this sense, we have tried again in a new paragraph (lines 186-196). We point out that our selection of 30 twin plots is significantly different (with higher mean similarity values) from the set of selections of 30 pairs that can be made randomly (1000 iterations) from our 60 sampled plots.
Second, in concerning the Method section parts and their related texts in the Results section, we have introduced two new subsections in the methods section with that intention.
Line 185. Validation of twin plot experimental design.
Line 197 Calculation of local and regional diversity
Regarding the other points you mentioned in the comments in your PDF, we have corrected them one by one following your suggestions, although we point out the most important modifications here.
Line 119 acidic by acid as you suggested.
Line 130. We have added a short phrase; “and lower densities of other herbivores such as wild boar (Sus scrofa L.) and roe deer (Capreolus capreolus L.)”, answering your question. Are there other ungulates or smaller herbivores which were also excluded from the sampling plots?
Line 191 we have eliminated the sentence because the results of the estimator have not been reported anyplace.
Lines 227 and 234 comments. We have added a new paragraph at the end of section methods explaining how the gross abundance is calculated, and how we used t-test for comparing, open and fenced plots, the statistical means of the two variables.
Line 265. We have corrected the sentence as you suggested.
Line 397. We have modified the paragraph explaining the validity of our group of twin plots.